# The Quantification of Spike Proteins in the Inactivated SARS-CoV-2 Vaccines of the Prototype, Delta, and Omicron Variants by LC–MS

**DOI:** 10.3390/vaccines11051002

**Published:** 2023-05-20

**Authors:** Kangwei Xu, Huang Sun, Kaiqin Wang, Yaru Quan, Zhizhong Qiao, Yaling Hu, Changgui Li

**Affiliations:** 1NHC Key Laboratory of Research on Quality and Standardization of Biotech Products, NMPA Key Laboratory for Quality Research and Evaluation of Biological Products, National Institutes for Food and Drug Control, No. 2, Tiantan Xili, Dongcheng District, Beijing 100050, China; xkwnifdc@163.com (K.X.);; 2Sinovac Life Sciences Co., Ltd., No. 21, Tianfu St., Daxing Biomedicine Industrial Base of Zhongguancun Science Park, Daxing District, Beijing 100050, China

**Keywords:** SARS-CoV-2, spike, standardization, LC–MS, vaccine

## Abstract

Developing variant vaccines or multivalent vaccines is a feasible way to address the epidemic as the SARS-CoV-2 variants of concern (VOCs) posed an increased risk to global public health. The spike protein of the SARS-CoV-2 virus was usually used as the main antigen in many types of vaccines to produce neutralizing antibodies against the virus. However, the spike (S) proteins of different variants were only differentiated by a few amino acids, making it difficult to obtain specific antibodies that can distinguish different VOCs, thereby challenging the accurate distinction and quantification of the variants using immunological methods such as ELISA. Here, we established a method based on LC–MS to quantify the S proteins in inactivated monovalent vaccines or trivalent vaccines (prototype, Delta, and Omicron strains). By analyzing the S protein sequences of the prototype, Delta, and Omicron strains, we identified peptides that were different and specific among the three strains and synthesized them as references. The synthetic peptides were isotopically labeled as internal targets. Quantitative analysis was performed by calculating the ratio between the reference and internal target. The verification results have shown that the method we established had good specificity, accuracy, and precision. This method can not only accurately quantify the inactivated monovalent vaccine but also could be applied to each strain in inactivated trivalent SARS-CoV-2 vaccines. Hence, the LC–MS method established in this study can be applied to the quality control of monovalent and multivalent SARS-CoV-2 variation vaccines. By enabling more accurate quantification, it will help to improve the protection of the vaccine to some extent.

## 1. Introduction

Since the emergence of the COVID-19 pandemic, its pathogen, a novel coronavirus (SARS-CoV-2), has continued to undergo genetic mutations [1,2,3,4,5]. SARS-CoV-2 is an enveloped, positive-sense, and single-stranded RNA viruse belonging to the Beta coronavirus genus of coronaviruses [6]. SARS-CoV-2 encodes 16 predicted non-structural proteins (NSP1–NSP16) and 4 structural proteins: spike (S), envelope (E), membrane (M), and nucleocapsid (N) proteins [7,8]. The S protein is a glycoprotein located on the surface of the coronavirus which can be cleaved by host cell serine protease TMPRSS2 at the S1/S2 cleavage site. The receptor binding domain (RBD) on the S1 subunit is able to bind to the ACE2 receptor on the surface of the host cells, allowing the virus to enter and infect the host [9]. Montenegro, R. C. et al. investigated the differences between coronaviruses, including variations in the S protein that make SARS-Cov-2 more virulent than other coronaviruses [10]. As the S protein is an essential neutralizing epitope for both inactivated and recombinant COVID-19 vaccines, the primary active ingredient is the S protein which is quantified using immunological techniques such as ELISA.

Vaccination is an effective means of preventing and controlling COVID-19. However, the continuous mutation of the virus has led to a reduction in the protective efficacy of the vaccines. The Delta variant, which was first detected in India in March 2021, quickly became the dominant global epidemic strain until it was displaced by the Omicron variant at the end of 2021. The Delta variant is known to be more transmissible and pathogenic than the prototype strain [11,12,13,14]. The Omicron strain has amassed more mutations than any other variant which makes it difficult for the body to eliminate the virus [15]. The protective efficacy of many existing vaccines against Omicron variants has been significantly reduced. Alex Sigal et al. found that the serum neutralizing antibody titer targeting Omicron of people previously vaccinated with the Pfizer BNT162b2 mRNA vaccine showed a significant decrease, with an average of a 41-fold decrease by being vaccinated alone [16]. Sandra Ciesek et al. investigated the effect of dosing regimens of mRNA vaccines on the neutralizing capacity against Omicron variants, including BNT162b2 (Pfizer, 2 dose), Spikevax (Moderna, 2 dose), and Vaxzevria (AstraZeneca, 1 dose) + BNT162b2 (Pfizer, 1 dose), and found that the serum antibodies from subjects vaccinated with three regimens cannot neutralize Omicron strain after 6 months. After an additional booster dose of BNT162b2, the serum neutralization rate only increased to 25% [17]. The protective efficacy of the inactivated vaccine was also substantially reduced. A total of 28 days after receiving the third dose of the inactivated vaccine (Sinopharm BIBP), 228 volunteers (78.08%) showed detectable serum neutralizing activity against the Omicron variant, with a 20.1-fold decrease in the geometric mean titer of neutralizing antibody compared to the prototype strain [18].

It becomes very necessary to have more effective vaccines to protect humans from Omicron or other variants. Developing broad-spectrum vaccines is a potential solution to the problem of reduced protective efficacy against VOCs [19,20,21,22,23]. Hwang et al. proposed the design of developing a universal vaccine (pan SARS-CoV-2). The inclusion of non-S protein structural proteins (nucleocapsid (N), member (M), and envelope (E)) in the target design of the vaccine aims to compensate for the T cell immune deficiency caused by Spike-only vaccine immunity. By triggering a strong cellular immune response, it synergizes with the B cell response, leading to the long-term success of the vaccine [24]. Furthermore, it has been reported that the frequencies of cytokine-secreting non-spike specific CD4+ and CD8+ T cells correlate well with those of the spike-specific T cellsfrom WT, Delta, and Omicron strains [25]. Despite partial broad-spectrum protection observed in animal studies, there are currently no broad-spectrum vaccines available on the market.

Another solution is the development of multivalent vaccines for vaccination. On one hand, it may avoid the poor production of neutralizing antibodies against subsequent variants due to the immune anchoring/antigenic guilt effect. On the other hand, recent researches have suggested that booster immunizations with bivalent vaccines including prototype and Beta strains could yield higher neutralizing antibody titers against Delta and Omicron variants than that with monovalent vaccines [26]. The phase II/III clinical trials of the Moderna bivalent mRNA vaccine consisting of the Prototype and Omicron strain as the booster showed 1.75 times higher levels of neutralizing antibodies against Omicron elicited by the bivalent one than the one induced by the monovalent one [27]. Pfizer and Moderna bivalent mRNA vaccines have been approved for emergency use by the FDA. Sinovac is developing a trivalent inactivated vaccine that includes the prototype strain as well as the Delta and Omicron variants.

For both inactivated and recombinant COVID-19 vaccines, the main target antigen is the S protein, of which the quantification is usually performed by using immunological methods such as ELISA. Due to the high similarity of S protein epitopes among different variants, it is difficult to effectively differentiate using antibodies. Therefore, the accurate quantification of each component cannot be achieved when evaluating the antigenic content of trivalent vaccines including the prototype, Delta, and Omicron strains. LC–MS/MS methods have also been reported for the quantification of S and N proteins. Zhen Long et al. and Lisa H et al. established LC–MS methods for the analysis of S and N proteins in inactivated vaccine stock solutions or patient specimens of the prototype strain, respectively [28,29]. Osnat R et al. used a similar method to evaluate the S protein content in a VSV-based vaccine [30]. Carrie P et al. developed an isotope dilution tandem mass spectrometer (IDMS) method for the quantitative detection targeting the N protein as well as the conserved sequence of the S proteins among prototype, Alpha, Beta, Gamma, and Delta strains [31]. However, studies on the detection of variant-based vaccines and multivalent vaccines have not been reported yet.

In this study, an IDMS method was developed for the quantification of S proteins among the prototype, Delta, and Omicron strains of the SARS-CoV-2 virus. This method could be used to quantify monovalent SARS-Cov-2 vaccines and it also could accurately distinguish and detect different S proteins from trivalent vaccines composed of the three strains above.

## 2. Methods

### 2.1. Selection and Synthesis of Native and Isotopically Labeled Strain Specific SARS-CoV-2 Peptides

The amino acid sequences of the S protein for the prototype, Delta, and Omicron strains of SARS-Cov-2 were digested using Biopharm Finder 4.1 software. The resulting computational peptides were compared and analyzed to identify a unique peptide sequence that could distinguish the S protein of all three strains. The selection of peptides is based on two principles: firstly, it must have strain specificity, that is, only one of the three strains has the peptide. It is best for peptides at the same location to have specificity for all three strains in order to avoid differences caused by the efficiency of trypsin digestion affected by the different S protein structures. Secondly, in order to obtain better MS response signals, it is best to select peptides that meet the general principles of MS detection peptide selection, such as the peptide length should be 7–20 amino acids. Extremely hydrophobic peptides should be avoided. Cysteine, methionine, and N-terminal glutamine amino acid residues should be avoided [32]. The final selected peptides are shown in Table 1. GenScript Biotechnology Co., Ltd. (Nanjing, China) was commissioned to synthesize both native and isotope-labeled (13C/15N Val on N terminus) peptides which were lyophilized and stored at −20 °C. The correctness of the synthesized peptide sequence was confirmed using Vanquish UHOLC-Q Exactive Plus Orbitrap Mass spectrometry.

### 2.2. Samples

Inactivated vaccine candidates and bulks for the prototype, Delta (B.1.617.2), and Omicron (B.1.1.529) strains of the SARS-CoV-2 virus were produced by Sinovac Biotech Ltd. (Beijing, China) following a previously reported process. Specifically, each virus strain was propagated in Vero cells and inactivated using β-propiolactone. The resulting three virus bulks were purified using ultrafiltration and chromatography purification. Finally, the trivalent vaccine was prepared by mixing and diluting the three virus bulks.

### 2.3. Preparation of Standard Solutions and the Samples

To prepare the standard solutions, six peptides (E, G, I, F, H, and J) were accurately weighed, dissolved in acetonitrile, and diluted with 50 mM ammonium bicarbonate solution to the following concentrations: E 400 ng/mL, G 400 ng/mL, I 500 ng/mL, F 360 ng/mL, H 300 ng/mL, and J 360 ng/mL. These six peptides were then proportionally mixed and diluted to each point concentration of the standard curve as shown in Table 2, using 50 mM ammonium bicarbonate.

For the internal standard polypeptide solution, three isotope-labeled peptide solutions were mixed proportionally and diluted with 50 mM ammonium bicarbonate. The final mixture contained F at 7.2 ng/mL, H at 6.0 ng/mL, and J at 7.2 ng/mL.

To prepare the test samples, 0.5 mL of each sample was taken and placed into a 1.5 mL microcentrifuge tube. The tube was then vacuum centrifuged at 60 °C until it was completely dry (about 2 h). Next, 100 μL of RapiGest solution with a concentration of 0.1% (086001861, Waters Corporation, Milford, MA, USA) was added to the microcentrifuge tube. RapiGest is an anionic surfactant that helps solubilize and unfold proteins making them more amenable to cleavage without denaturing or inhibiting common proteolytic enzymes, thereby reducing the digestion time. It is prone to hydrolysis in acidic environments to produce insoluble precipitates, thus not interfering with MS detection. The contents were vigorously vortexed to ensure complete mixing and then heated at 60 °C for 30 min to denature the protein and facilitate trypsinization. After cooling to room temperature, 0.5 mL of the above mixed internal standard polypeptide solution and 5 μL of trypsin solution at a concentration of 0.5 μg/μL (V5117, Promega, Madison, WI, USA) were added to the sample. The sample was then subjected to shaking digestion at 37 °C and 1000 rpm/min for 16 h. The protein was then hydrolyzed into peptides for MS detection. Then, 10 μL of trifluoroacetic acid was added and mixed well and left at 37 °C for 30 min to stop trypsin activity, thus reducing the pH of the solution to hydrolyze RapiGest. Finally, the solution was transferred into a 1 mL volumetric flask and brought to volume by 50 mM ammonium bicarbonate solution. After centrifuging at 11,000 rpm for 10 min, RapiGest hydrolysis products and other insoluble substances were removed. The supernatant was collected as the test solution.

### 2.4. LC/MS/MS Instrumentation Parameters

The Vanquish UHPLC-TSQ Quantis liquid chromatography–triple quadrupole mass spectrometry (Thermo Scientific, Waltham, MA, USA) was used to separate and detect the peptides. The instrument is subjected to annual statutory testing annually to ensure its performance meets requirements. Pierce Triple Quadrupole Calibration Solution Extended Mass Range (Thermo Scientific, Waltham, MA, USA) was used for calibration. The analytical column, Shim-pack GISS-HP C18 Metal-free (2.1 × 150 mm (HSS), 3 μm (SHIMADZU) was maintained at 35 °C. The mobile phase consisted of 0.1% formic acid in acetonitrile (mobile phase A) and 0.1% formic acid in water (mobile phase B). Samples of 40 μL were injected into the analytical column and the chromatographic separation gradient was performed at a flow rate of 0.3 mL/min according to the following conditions: initial, 95% B; 1 min, 95% B; 8 min, 60% B; 8.1 min, 10% B; 10 min, 10% B; 10.1 min, 95% B; and 15 min, 95% B. The total running time was 15 min.

The TSQ system, equipped with an ESI interface operating in the positive ion MRM mode, was used to analyze the column eluent. The precursor/product pairs and collision energies for peptides were shown in Table 1. The instrument parameters were set as follows: spray voltage 3800 V, sheath gas 50 arb, auxiliary gas 15 arb, ion Transfer Tube Temperature 350 °C, and CID Gas 2 mTorr. The collision energy and tube lens of each peptide were optimized separately. Thermo Scientific Chameleon7.2 was used for instrument control and data analysis.

### 2.5. Data Analysis

To enhance the accuracy and precision of the quantitative method, internal standard peptides were employed using the internal standard method. Isotope-labeled peptides with the same concentration as the internal standards were added to all standard and sample solutions. The area ratio of the target peptide to the isotope-labeled peptide was subjected to linear fitting against the concentration of the target peptide to establish a standard curve. This curve was then utilized to determine the content of the peptides in the S protein. The S protein content (pmol/mL) in the samples was calculated using the following formula: $S\;\mathrm{protein}\;\mathrm{content}\;\left(\mathrm{pmol}/\mathrm{mL}\right)=\frac{\mathrm{peptide}\;\mathrm{content}\;\left(\mathrm{ng}/\mathrm{mL}\right)}{\mathrm{molecular}\;\mathrm{weight}\;\times 1000}$.

## 3. Results

### 3.1. Select of Strain Specific SARS-CoV-2 Peptides

Trypsin digestion was simulated after aligning the S proteins of three strains (prototype, Delta, and Omicron strains) and the peptide sequences after digestion were compared. We identified two sets of peptides that can be used to distinguish the three variants at the same time (Figure 1). The first group is located at positions 445–454, with the peptide sequences VGGNYNYLYR, VGGNYNYR, and VSGNYNYLYR, for the prototype, Delta, and Omicron strains, respectively. The second group is located at positions 130–147 with the peptide sequences VCEFQFCNDPFLGVYYHK, VCEFQFCNDPFLDVYYHK, and VCEFQFCNDPFL---DHK, for the three strains, respectively. The first set of distinct peptides was finally selected for quantitative studies for the following reasons: firstly, the second group of peptides contained several disulfide bonds, which could affect the stability of results. Secondly, the first group of peptides is shorter in length (the longest peptide is only 10 amino acids) compared to the second group (the longest peptide is 18 amino acids), resulting in a better mass spectrometry response. Lastly, some samples could not be detected using the second group polypeptides.

### 3.2. Method Validation

The method was validated for specificity, linearity, recovery, and precision.

#### 3.2.1. Specificity

The monovalent bulks of the prototype, Delta, and Omicron strains were taken and treated according to the method described in Section 2.3 without adding any isotope-labeled peptides. The results are presented in Figure 2, which showed that only the strain-specific peptides were detected in each bulk. This demonstrated that using the strain-specific peptides can effectively distinguish the three strains and confirmed that the source of the peptides was the S protein in the corresponding strain rather than other matrix proteins, indicating specificity and anti-matrix interference ability.

#### 3.2.2. Linearity

The standard curves were generated using different serial concentrations of the peptides. Linear regression was used to fit the peptide concentration versus the area ratio relationship. As shown in Figure 3, all three peptides showed a strong linear relationship with a regression correlation coefficient R^2^ greater than 0.99.

#### 3.2.3. Recovery

To assess the matrix effect, peptide control solutions with different concentrations were added to the trivalent vaccine samples which were processed and detected simultaneously with the samples without the added peptides. The recovery rate was calculated using the formula: $\mathrm{recovery}\left(\%\right)=\frac{\mathrm{sample}\;\mathrm{with}\;\mathrm{added}\;\mathrm{peptide}-\mathrm{sample}\;\mathrm{without}\;\mathrm{peptide}}{\mathrm{theoretical}\;\mathrm{peptide}\;\mathrm{addition}\;\mathrm{amount}}\times 100\%$. The average recovery rates of the prototype strain (peptide E), Delta strain (peptide G), and Omicron strain (peptide I) were 112.81%, 94.07%, and 87.71%, respectively. These results indicated that the method has good accuracy and the matrix has little influence on the method.

#### 3.2.4. Precision

Repeatability was evaluated by processing a batch of trivalent vaccine samples six times, yielding RSDs of 5.5%, 5.7%, and 5.8% for peptides E, G, and H, respectively. The intermediate precision was assessed by analyzing the same batch 12 times over 2 days by 2 experimenters, yielding RSDs of 5.0%, 6.1%, and 6.8% for peptides E, G, and H, respectively. Both the repeatability and intermediate precision demonstrated that RSDs were within 10%, indicating good precision of the new method.

### 3.3. Quantification of S Protein in Inactivated Vaccine Bulks and Trivalent Vaccines

The content of S protein in vaccine bulks was quantified using the validated method. In total, 18 batches of vaccine bulks from prototype, Delta, or Omicron strains were analyzed individually. The results are shown in Table 3. Data showed that the S protein content was consistent within different batches of the same strain but varied greatly among three different strains, with the prototype bulk having the highest S protein content. The average S protein content of the Delta and Omicron bulk solutions was approximately 1/10 and 1/7 that of the prototype, respectively. Previous studies have reported that SARS-CoV-2 virus, produced in Vero cells, was prone to mutations in S1/S2 deletion which enhanced growth and stability in vitro [33]. ΔS679–688 has also beenfound in inactivated prototype strain vaccines, located at the S1/S2 junction [34]. However, no similar mutations were observed in vaccines produced with the Delta or Omicron strain. It was suggested that S1/S2 deficiency may increase the proportion of S protein, which is consistent with data observed in Table 3.

Three batches of trivalent vaccines were produced by mixing different strain bulks, with the same S protein final concentration of 2.5 pmol/mL for three strains. The S protein content of each trivalent sample was measured and the results are shown in Table 3. The content of each strain ranged from 2.3 to 2.7 pmol/mL, in general agreement with the theoretical proportion, confirming the accuracy of the method for trivalent samples. These results further verified the specificity and accuracy of the method.

## 4. Discussion

The S protein is the major effective antigenic component of inactivated SARS-CoV-2 vaccine and its accurate determination is a key indicator for vaccine quality control, which is usually quantitatively detected by immunological methods such as ELISA. However, due to the high homology of S proteins among the various SARS-CoV-2 variant, methods based on antibody–antigen interactions are difficultly applied to the identification of different variants, as well as to the S protein quantification of different strains in multivalent vaccines. The results published by BIBP showed that the ELISA method for S protein of the prototype, Delta, and Omicron strains had great interference in detecting bivalent or trivalent vaccines composed by the three strains. The interference rate generated by this ELISA method displayed that the Delta/Omicron versus prototype was 6.49%, the prototype/Omicron versus is 140.7%, and the Delta/prototype versus Omicron was 184.41% in the trivalent vaccine, in which the ratio was 1:1:1 by three strains. So, it is difficult to accurately quantify the monovalent components in trivalent vaccine [35]. The ELISA method used by Sinovac had a similar situation (data unpublished). In the previous study of neuraminidase content detection of trivalent influenza vaccine, our laboratory prepared type-specific antibodies for specific peptides (15 amino acids) in N1, N2, and NB and established a Slot Blot detection method using these antibodies [36]. Although many different sites of amino acid sequences were observed in the strains between genotype, Delta, and Omicron, no specific peptides with consecutive multiple amino acids were found. Theoretically, the specific antibodies for mutant-site epitopes can largely avoid the cross-reactivity of S proteins between different strains, but so far there is no report of an immunological method that can precisely distinguish the S proteins of three strains.

Using molecular biology methods to identify SARS-CoV-2 variants was generally applied in surveillance and clinical diagnosis. Deep sequencing technology is widely utilized to identify SARS-CoV-2 variants, which can identify each mutation in the sample [37,38]. Real-time PCR assays for identifying SARS-CoV-2 variants were also reported [39,40]. While these molecular biology methods could be used to identify inactivated SARS-CoV-2 vaccines, the RNA level of vaccines may not necessarily be correlated to the S protein content.

MS methods offer high specificity, sensitivity, and accuracy for qualitative and quantitative analysis of complex biological samples. It is widely used in biomedical fields such as clinical diagnosis, drug research, and development [32,41]. It is also used in the development and quality control of HPV, Ebola, HIV, influenza, SARS-CoV-2, and other vaccines [29,30,31,42]. Among the many mass spectrometry quantification methods, isotope dilution tandem mass spectrometer (IDMS) has been recognized as the reference method [43]. The IDMS method is typically carried out using the following steps. The proteins that need to be quantified are enzymatically digested into peptides and specific peptides are selected for quantification. A fixed amount of synthetic 13C and 15N isotopically labeled peptides are spiked into the standards and samples as internal standards. After enzymatic digestion, samples are separated by liquid chromatography and detected by MS in multiple reaction monitoring modes. The area ratio of the target peptide and isotope-labeled peptide is linearly fitted to the concentration of the target peptide to obtain a standard curve, which was used to quantify the content of protein. Since isotopically labeled peptides are simultaneously processed with the sample for digestion and LC–MS detection, quantified by area ratio, the influence of the pretreatment and injection volume on the results can be reduced, allowing for accurate and precise quantification [31,43]. Tracie L et al. have established the IDMS method for quantifying influenza vaccine HA and other proteins, which has been applied in the evaluation of influenza vaccine strain products and the development of international standards [44]. A similar IDMS method for the quantitative detection targeting the S protein has also been established [31]. However, studies on the detection of variant-based vaccines and multivalent vaccines have not been reported yet.

Here, we established an IDMS method for quantifying S protein by targeting specific peptides that can simultaneously distinguish the three variants. This method has good specificity. In monovalent stocks, only the signal of strain-specific peptides can be detected rather than other strain or matrix proteins (Figure 2). Furthermore, the results of S protein content detection was consistent with the theoretical values in the trivalent vaccine mixed with three virus bulks (Table 3), indicating that there was no significant mutual interference among the three strains.

Although the IDMS method enables the precise and accurate quantification of S proteins, there are some limitations that require careful discussion. As an antigen, the S protein plays a critical role in stimulating the production of neutralizing antibodies that protect against infection. The use of neutralizing antibodies to establish the ELISA detection method can reflect the integrity of protein antigen structure, which is also an important reason why many vaccines use ELISA method for antigen content detection [42,45,46]. Unlike the ELISA method, MS directly detects the polypeptide after enzymatic digestion of proteins, which has the same number of moles as the protein, to calculate the protein content. Most epitopes are destroyed during enzymatic digestion of proteins, so the results obtained by the MS method may not reflect the destruction of epitopes. Therefore, for vaccine stock solutions and monovalent vaccines that can be quantified by the ELISA method, we do not recommend using this method to replace the ELISA method. Rather, as a supplementary method for better quality control of vaccines or for the preparation and assignment of standards. For the inactivated trivalent SARS-CoV-2 vaccine, from what we know, the method established in this study is currently the only accurate option for quantification.

## 5. Conclusions

In this study, a novel method was developed for the quantification of the S protein in monovalent or trivalent inactivated vaccines against the SARS-CoV-2 virus (prototype, Delta, and Omicron strains). Compared with the traditional methods, such as ELISA, this method can identify and quantify the S protein in different strains of the SARS-CoV-2 virus (prototype, Delta, and Omicron strains), as well as compare the S protein content in different batches of trivalent vaccines and assess the batch-to-batch stability, which can provide valuable reference points for establishing product quality control standards. The high selectivity of liquid chromatography–mass spectrometry (LC–MS) ensures minimal interference from complex sample matrices, making the method useful for the quality control of monovalent or trivalent inactivated vaccines.

Although there are many reports on SARS-CoV-2 non-spike proteins, their roles mostly remain unclear [29]. We believe that further LC–MS/MS methods could extend the quantification of these proteins when researchers grasp their characteristics deeply and understand their impact on the safety and efficacy of vaccines. In the event that a new variant emerges that cannot be distinguished by existing target peptides, accurate quantification could still be achieved by synthesizing new target peptides and isotopically labeled peptides within a few weeks, as described previously.

## Figures and Tables

**Figure 1 vaccines-11-01002-f001:**
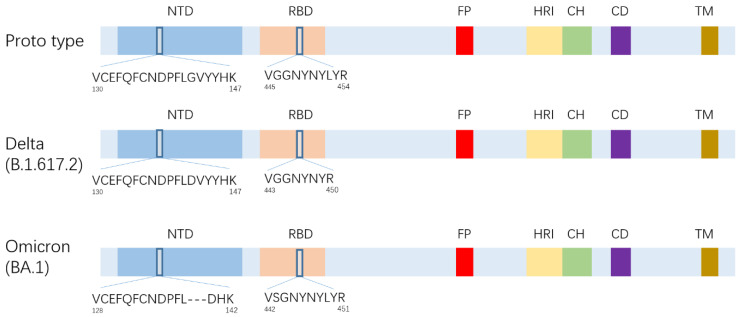
Schematic diagram of strain-specific SARS-CoV-2 peptide selection. There were two sets of peptides at the same location to have specificity for all three strains which were located in the RBD region at positions 445–454 and NTD at positions 130–147. The 445–454 position peptides were more suitable for MS detection, so this group of peptides was selected for method establishment.

**Figure 2 vaccines-11-01002-f002:**
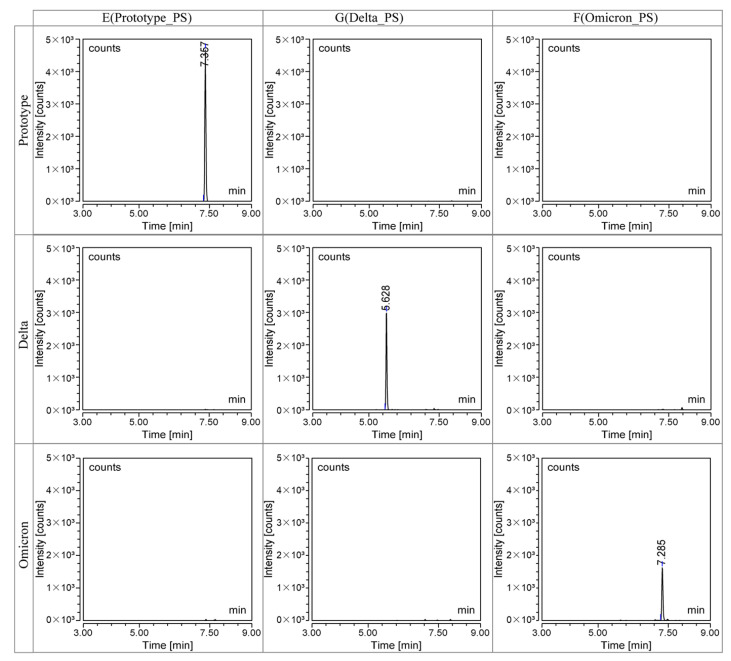
Specific characteristic spectrums of different variants. Monovalent bulks of the prototype, Delta, and Omicron strains were analyzed by LC–MS method. In the prototype bulk (**first line**), only peptide E can be detected. In the Delta bulk (**second line**), only peptide G can be detected. In the Omicron bulk (**third line**), only peptide I can be detected. The selected peptides have good strain specificity and can only be detected in the corresponding strains and do not exist in other strains or matrices.

**Figure 3 vaccines-11-01002-f003:**
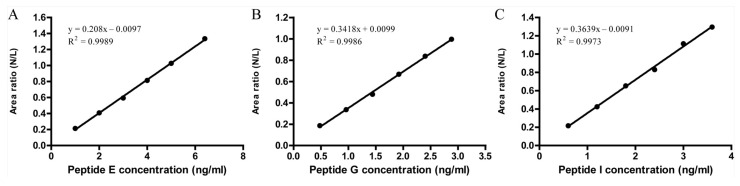
Standard curve for three peptide standards showing linearity. The X axis indicates the concentration of the target peptide and the Y axis indicates the area ratio of the target peptide versus the isotope-labeled peptide. The standard curve is obtained after linear fitting. The standard curves of all three viruses have good linearity with R^2^ > 0.99. (Panel **A**) is the standard curve for the prototype strain, (Panel **B**) for the Delta strain, and (Panel **C**) for the Omicron strain.

**Table 1 vaccines-11-01002-t001:** Strain specific peptides of the SARS-CoV-2 S protein.

Variant Strain	Peptide Name	Peptide Sequence	Precursor Ion *m*/*z*	Target Ion *m*/*z*	Collision Energy (%)
Prototype strain	E	VGGNYNYLYR	610.08 (+2)	1119.41	23
	F	**VGGNYNYLYR**	613.08 (+2)	1119.58	23
Delta (B.1.617.2)	G	VGGNYNYR	471.98 (+2)	843.33	16
	H	**VGGNYNYR**	474.56 (+2)	843.25	18
Omicron (BA.1)	I	VSGNYNYLYR	625.05 (+2)	728.58	20
	J	**VSGNYNYLYR**	628.03 (+2)	728.5	21

Underlined amino acids indicate the 13C/15N isotope labeled.

**Table 2 vaccines-11-01002-t002:** Serial concentrations of standard polypeptides.

Standard Polypeptides			Concentration	(ng/mL)		
Working Solution	1	2	3	4	5	6
polypeptide E	1.0	2.0	3.0	4.0	5.0	6.4
isotope-labeled peptide F	3.6	3.6	3.6	3.6	3.6	3.6
polypeptide G	0.48	0.96	1.44	1.92	2.4	2.88
isotope-labeled peptide H	3.0	3.0	3.0	3.0	3.0	3.0
polypeptide I	0.6	1.2	1.8	2.4	3	3.6
isotope-labeled peptide J	3.6	3.6	3.6	3.6	3.6	3.6

**Table 3 vaccines-11-01002-t003:** S protein content in inactivated SARS-CoV-2 vaccine bulks and trivalent vaccines.

Samples	Lot	Spike Protein Concentration (pmol/mL)		
		Prototype Type	Delta	Omicron
Vaccine bulks of the prototype strain	1	68.5	/	/
	2	68.7	/	/
	3	67.7	/	/
	4	68.3	/	/
	5	71.3	/	/
	6	72.3	/	/
Vaccine bulks of the Delta strain	1	/	7.6	/
	2	/	7.6	/
	3	/	5.6	/
	4	/	5.8	/
	5	/	7.9	/
	6	/	5.1	/
Vaccine bulks of the Omicron strain	1	/	/	10.3
	2	/	/	11.7
	3	/	/	10.3
	4	/	/	9.2
	5	/	/	9
	6	/	/	10.3
trivalent inactivated vaccine	1	2.6	2.5	2.4
	2	2.6	2.4	2.6
	3	2.7	2.6	2.3

“/” indicate not detected.

## Data Availability

Data will be made available on request.

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
