# Peer review of "The Quantification of Spike Proteins in the Inactivated SARS-CoV-2 Vaccines of the Prototype, Delta, and Omicron Variants by LC–MS"

_vaccines, 2023, doi:10.3390/vaccines11051002_

Round 1

Reviewer 1 Report

Using a unique approach, the scientists claim to be able to identify and quantify the S protein in several strains of the SARS-CoV-2 virus, including the Delta and Omicron strains, in monovalent or trivalent inactivated vaccines. Although this is of great interest, I believe this work has some issues that need to be fixed before the reported approach can be widely accepted and believed. I'm hoping the authors will heed these suggestions for improvement and present more solid, carefully vetted facts in future works. I have a few suggestions for the author to think about:

 1.      The abstract needs a stronger introductory sentence that more clearly states the issue being investigated. It would be more informative to highlight the specific obstacle that the work is addressing, such as the inability to reliably distinguish and quantify SARS-CoV-2 variations using immunological approaches, rather than asserting that producing variant vaccines is doable.

2.      The prospective consequences of the study may be stated more clearly in the abstract. The authors may, for instance, advocate more concrete applications for the LC-MS method in future vaccine development efforts or emphasize the need for improved techniques for measuring SARS-CoV-2 variations in vaccine studies.

3.      The introduction might use a more organized and logical layout. There is a lack of coherence in the presentation of the data; many topics and studies are brought up without much rhyme or reason. Assembling similar pieces of data and clearly denoting shifts between topics would improve efficiency.

4.      A more analytical approach to the introduction's writing is warranted. While the research cited does fill a gap in our knowledge, this section does not address the studies' caveats or potential biases. There is no consideration of potential confounding factors or alternative interpretations of results, such as the considerable reduction in the serum neutralizing antibody titer targeting Omicron following vaccination.

5.      The methodology section is short in explaining why these particular peptides were chosen and what makes them special. Some context and justification for the choice of these peptides would be helpful.

6.      Insufficient information about how to prepare samples for testing is provided in the methods section. For instance, the function and rationale behind the incorporation of RapiGest solution into the preparation procedure remain mysterious. Readers would have a deeper understanding of the methodology if a greater explanation was provided for the decisions that were made.

7.      The LC/MS/MS instrumentation specifications are given, but it would be great to know more about the instrument's calibration and the methods used to interpret the data. It would also be helpful to know what sort of quality control methods were implemented during the trial.

8.      Additional methodological specifics, such as trypsin digestion conditions, peptide identification, and mass spectrometry processes, would enrich the results section. This would help the reader understand the study's significance and make future replications less cumbersome.

9.      There is room for improvement in the results discussion, particularly with regards to the ramifications of the findings. Examples of topics that could be covered in this part include the limits of the current approach, potential applications of the identified strain-specific peptides in vaccine formulation and monitoring, and suggestions for future studies.

10.   More visual aids would help convey the information in this section. Including summaries of the results in the form of tables or graphs, for instance, would facilitate analysis and comparison. For the benefit of the reader who is not as well versed with the methodologies, it may be useful to include pictures or diagrams of the peptides or the experimental apparatus.

11.   More background on why measuring the S protein in different viral strains is so crucial, especially in light of the introduction of new variants like Delta and Omicron, would be helpful in the conclusion section. Readers would benefit from this background information since it would help them place the study's findings in the context of the present worldwide health problem.

12.   It would be helpful if the limits of the refined LC-MS approach were more explicitly stated in the conclusion section. Is precise quantification still difficult for any particular sample matrix or condition? Readers would have a deeper understanding of the method's potential use if these restrictions were addressed.

13.   The authors say they can detect and measure the S protein in all known SARS-CoV-2 strains, including Delta and Omicron. There is no proof that the S protein can be detected or quantified using their method in these strains, which all have unique mutations that may alter its structure and function. The authors also fail to account for the likelihood that additional proteins or impurities present in the vaccine samples will impede the LC-MS analysis, reducing the reliability of their findings.

14.   The authors propose their method for evaluating trivalent vaccinations' consistency from batch to batch and for setting quality assurance benchmarks. They do not, however, include information about the consistency and reliability of their procedure or the variation in S protein level between batches of vaccines. In the absence of such data, it would be premature to assert that their approach can be utilized for quality control.

Reviewer 2 Report

In this paper Xu et al described a LC-MS method to quantify the S proteins in inactivated monovalent vaccine or trivalent vaccine (prototype, Delta and Omicron strains). First of all, they identified small peptides from S protein specific among the three strains (prototype, Delta and Omicron), then they synthesized the corresponding labelled peptides and used them as references in LC-MS analysis. Quantitative analysis was performed by calculating the ratio between the reference and internal target.

The idea of finding a way to quantify proteins is valid but in this paper the results have not been well explained and commented. From the reported data it is not clear how the peptides were obtained and how the presence of the labeled peptides can be exploited for the determination of the proteins content. The method has not been compared with the measurements coming from other experiments (different techniques) but above all it is not clear how the conclusions were reached from the data obtained.

Tables 3 and 4 and the figures reported need to be better commented in the text.

Therefore my suggestion is to better rewrite the results chapter to make the reader understand which data have been obtained and their meaning. Furthermore, the values obtained must be compared with those known in the literature or obtained with different techniques.

English language is fine, please check some spelling 

Author Response

We sincerely thank reviewer for thorough revision and detailed suggestions for our MS. The following is our point-to-point revision or reply according to your suggestions.
1.    In order to make the results clearer, we have made the following changes:
Rewrite results section 3.3.
Added discussion section to further analyze the results. 
Added section 2.5 “Data analysis” in the method section to explain how the results were obtained. 
Tables 3 and 4 were modified and merged into table 3. 
Added content of the figure legend.
2.    For the method comparation:Firstly, we conducted detailed validation on the specificity, precision, accuracy, and other aspects of the established method to ensure its accuracy and reliability as much as possible. Due to its pioneering nature, there is no available method for comparison. In the discussion, we also described the situation that the poor specificity of existing ELISA methods.
3.    The peptide source can be found in Method 2.1, and the GenScript Biotechnology Co is commissioned to synthesize it.

Reviewer 3 Report

The submitted manuscript is well written and sounds interesting.

The text reports a method for the laboratory evaluation of the potential efficacy of different vaccines, and could be used for new vaccines other than for COVID-19 prevention in the future.

I've only a suggestion. It seems to be useful to expand the introduction with more informations about SARS-CoV-2 and immune response. If you agree, please read and discuss the following references:

doi: 10.1016/j.ijbiomac.2021.02.203.

doi: 10.1007/s00109-020-02012-8.

doi: 10.3389/fimmu.2023.1139620.

doi: 10.1371/journal.ppat.1010870.

English grammar and language require are fine, but a revision is useful for typos (i.e., l. 49: "does" is dose, etc.)

Author Response

We sincerely thank reviewer for thorough revision and detailed suggestions for our MS. The suggested references you mentioned in the letter are very helpful, and we incorporated them into the introduction section. We also revised the typos in the manuscript.

Round 2

Reviewer 1 Report

No further comments

Reviewer 2 Report

The authors answered to my comments and suggestions